# Evaluation of Size of Trunk Asymmetry in Children Practicing Selected Sports Disciplines

**DOI:** 10.3390/ijerph20064855

**Published:** 2023-03-09

**Authors:** Natalia Twarowska-Grybalow, Aleksandra Truszczyńska-Baszak

**Affiliations:** Department of Physiotherapy, Józef Piłsudski University of Physical Education in Warsaw, 00-968 Warszawa, Poland; natalia.twarowska@gmail.com

**Keywords:** body posture, Moiré method, physical activity, spine

## Abstract

(1) Background: The aim of the study was to assess the body posture of children practicing selected sports disciplines and to compare it to the body posture of non-training children. (2) Methods: 247 children practicing a selected discipline either in primary sports schools or in sports clubs constituted the study group. The control group was composed of 63 children that did not practice any sport. The study of body posture by using the Moiré method allowed for assessing the size of parameters determining body posture. Selected parameters characterizing the position of the shoulders and shoulder blades, the waist triangle, and the position of the posterior iliac spines were analyzed. (3) Results: The differences in the selected parameters were not statistically significant in all parameters except the model describing the values determining the depth of the shoulder blades that were measured in millimeters between the groups. (4) Conclusions: Most of the examined people had correct body posture in the sagittal plane, regardless of the type of sport practiced. In all the examined groups, the most common dysfunctions were asymmetries of moderate intensity in the frontal plane. The results of our own research did not allow us to clearly state whether practicing different sports disciplines and different training loads has a negative or positive impact on body posture. The lack of asymmetry of high intensity in the groups of people practicing various sports disciplines, despite the fact that the given disciplines are asymmetric, may indicate that exercises during the training process are correctly selected.

## 1. Introduction

The slight curvature of the spine, the symmetrical arrangement of selected bone points and joints such as shoulders, shoulder blades, waist triangles, front and rear iliac spines, knees, and feet in relation to each other are all the necessary elements that should be taken into consideration when assessing if the body posture is correct. The correct posture can be also characterized by symmetrical positioning of the head in relation to the long axis, the slight arch of the chest, the same length of the upper and lower limbs, and the correct arch of the feet [1,2]. Correct body posture defines the optimal work of muscles and internal organs as well as neuromuscular coordination [3,4].

Posture defects in children and adolescents are a common phenomenon that currently affects from 30 to 60% of the population. They may involve the spine, trunk, chest, pelvis, or limbs [5,6,7,8]. For linear parameters, a difference in the arrangement of individual points in relation to each other exceeding 10 mm, indicates a significant asymmetry, whereas a difference between 5 mm and 10 mm means moderate asymmetry, and values below 5 mm indicate a lack of asymmetry. For the angle parameters, it was assumed that values above 3° mean significant asymmetry, values between 1.5° and 3° mean moderate asymmetry, and values below 1.5° mean there is no asymmetry [9,10,11,12].

Many authors [5,13,14] emphasized that the lack of prophylaxis and comprehensive correction of incorrect body posture may lead to the rapid progression of postural defects in adolescence. Spontaneous and accurate posture correction without the help of a specialist is a rarely observed phenomenon [5].

The review of the current literature did not give a clear answer to whether the professional practice of various sports disciplines has a positive effect on body posture, or whether it increases asymmetries and the body posture disorders of training athletes [15,16,17,18]. Each sports discipline is characterized by a different specificity, where the differences also apply to the training plan and the impact of training on the athlete’s body. Many publications have confirmed the positive effect of practicing sports on body posture [17,18]. On the other hand, researchers reported negative consequences of asymmetric physical activity on various elements of body posture [19,20,21,22,23]. The current state of knowledge validates also that there is an increase in the percentage of postural defects in children and adolescents in the last decade in spite of physical activity practiced by them [3,13,14,24,25,26].

Sports disciplines that increase the risk of postural asymmetry include: among others volleyball, team sports, and combat sports [15,16,20].

There are many studies confirming the incorrect type of posture in children practicing volleyball. Vařeková et al. [22] observed asymmetry in the position of the shoulder, scapula, and posterior superior iliac spine on the dominant side in professional players. Kuczyński et al. [27] presented a different theory showing that practicing volleyball results in better body posture stability and a different mode of automatic posture control in training children compared to the control group.

In recent years, the impact of martial arts training on body posture has also been analyzed. Wałaszek et al. [28] assessed body posture changes in a group of 6-year-olds practicing judo. Sunghak et al. [29] also showed a positive effect of martial arts on body posture correction.

The projection method with the Moiré system is a method of assessing body posture that supplements the X-ray examination of the spine. The Moiré system test is safe and does not expose the test participant to any negative health effects. The credibility of the research method using the Moiré system has been confirmed in recent years by numerous publications [11,30,31,32].

In this study, we posed the following questions: (1) What body posture disorders occur most frequently in the groups of training and non-training children? (2) How often do they occur? (3) What types of body posture disorders are characteristic of selected sports disciplines? The aim of the study was to assess the body posture of children practicing selected sports disciplines and to compare it to the body posture of non-training children.

## 2. Material and Methods

This research is an observational, case-control study. A total of 247 children practicing a selected discipline either in primary sports schools or in sports clubs constituted the study group. In the present study, the investigations described in the former work entitled “The Sizes of Spine Curvatures of Children That Practice Selected Sports” [33] were continued. For our analysis, we selected the following disciplines: biathlon/taekwondo, football, volleyball, and swimming. Children that trained in biathlon and taekwondo were pupils of the same class in a primary sports-profile school. The control group was composed of 63 children that did not practice any sport. All participants attended primary schools and sports clubs in Warsaw, Poland. The study was conducted in four primary schools and two sports clubs.

In the present study, the following inclusion criteria were used:(a)obtaining the written consent of the parent or legal guardian to conduct the examination;(b)the intensity of practicing a selected sports discipline—minimum 6 h a week (in the case of the study group);(c)age interval of the children was from 8.5 to 13.5 years;

Whereas the exclusion criteria were:(a)serious orthopedic injuries precluding equal load on both lower limbs;(b)structural scoliosis (that was excluded by the physical examination);(c)asymmetry of the length of the lower limbs higher than 2 cm (that was excluded by the physical examination);(d)other diseases that lead to musculoskeletal deformities or structural postural defects;(e)lack of written consent of the parent or legal guardian to conduct the examination.

In order to study the body posture we used the Moiré method which is in line with the guidelines of the device’s manufacturer [34]. Selected parameters characterizing the position of the shoulders and shoulder blades (KLB, OL, UL, UB), the waist triangle (TT, TS), and the position of the posterior iliac spines (KNM, KSM) were analyzed. The above parameters—which are marked in Figure 1—describe:KLB—the angle of inclination of the right and left shoulder;UL—the difference in height between the right and left shoulder blades;UB—the difference in depth between the right and left shoulder blades;OL—the difference in the distance between the right and left shoulder blades in relation to the spine line;TT—the difference in height of the right and left waist triangle;TS—the difference in width between the right and left waist triangle;KNM—the angle of inclination of the posterior iliac spines in the pelvis;KSM—the angle of twisting of the posterior iliac spines in the pelvis.

The parameters listed above can be calculated by marking the appropriate points on the body of the examined child. In this study, we were marking the spinous processes of all vertebrae of the spine, the cervicothoracic and thoracolumbar junctions, the peak of kyphosis and lordosis, and the sacrum. In addition, the inferior angle of the scapula, the posterior iliac spines, and the greatest depression in the waist were marked. The test was conducted in a darkened room, and the examined person was positioned perpendicularly and at a distance of 2.6 m from the equipment. The photograms were evaluated after completion of the examination, without the participation of the examined person.

Anthropometric measurements were performed and anthropological indicators were calculated for each participant of the study. Body weight was determined with an accuracy of 0.1 kg by using an electronic scale. Body height was measured with an anthropometer with an accuracy of 1 cm. BMI was calculated for each child by using data on weight and height. BMI percentage charts were calculated based on the OLAF and OLA project carried out by the “Pomnik—Centrum Zdrowia Dziecka” Institute in the years 2007–2012 [35].

The study was conducted in 2017–2020. The tests were carried out during morning school hours in the morning or on a date that was pre-determined with the coaches of sports clubs “Sparta” and “Escola Varsovia”. The collection of results was interrupted by the COVID-19 pandemic, which contributed to the closure of schools and the interruption of the training process in sports clubs. The data were collected by one researcher (NTG) who, after completing the methodological part, processed the results in accordance with the recommendations of the device’s manufacturer [34].

The number and type of physical activity in children in the control group were specified by using an original questionnaire.

### Statistical Methods

Simple descriptive statistics were initially calculated. For qualitative variables, the χ^2^ test or Fisher’s exact test was used, depending on the sample size. For quantitative variables, the hypothesis of equality of distributions was tested with the Kruskal–Wallis test. Spearman’s correlations were used to examine the relationships between the studied variables.

The final analysis of the results was based on multidimensional statistical GLM (Generalized Linear Models) models. According to the Akaike Information Criterion (AIC), the most appropriate model parameters were selected.

To assess the repeatability of the measurements, repeated observations of an independent group of patients were analyzed and then the ICC coefficient was calculated in a random effects ANOVA model.

The calculations were made in the SAS package (SAS/STAT rel. 15.1). The value of *p* = 0.05 was assumed as a significant level of confidence. In calculations that were statistically significant, the power of the test was shown to be valid.

## 3. Results

The children examined were aged from 8.5 to 13.5. The detailed biometric data of the examined children are presented in Table 1.

The conducted study allowed for a quantitative analysis of the training load in groups practicing selected sports disciplines. The detailed data on the number of training hours per week and the number of months since the moment when the training started are presented in Table 2.

The study of the body posture by using the Moiré method allowed for assessing the size of parameters determining the body posture.

Average values of the parameter describing the inclination of the right and left shoulder (KLB), defined in millimeters and degrees, in the GLM model were compared. There were no statistically significant differences in the above parameter between the study groups. The highest average values were presented by children training in biathlon/taekwondo. The lowest values were presented by children practicing volleyball and by children in the control group. All average values were within the range of moderate asymmetry. The detailed data are collected in Table 3.

Data on the parameters describing the differences in the height, depth, and distance of the blades (UL, UB, OL) in the GLM model were also collected and they were measured in millimeters and degrees. The characteristics of the parameters are presented in Table 4.

The difference in height between the right and left shoulder blades showed a moderate asymmetry in all the groups studied. The highest values of the parameter occurred in the control group. The differences between the groups were not statistically significant. The analysis of the depth between the right and left shoulder blades measured in millimeters showed a moderate asymmetry in the biathlon/taekwondo group and no asymmetry in other groups. The model describing the values determining the depth of the shoulder blades that were measured in millimeters showed statistically significant differences between the examined groups. The statistical model showed a difference between the biathlon and taekwondo group and the control group (*p* = 0.0067) and the swimming group (*p* = 0.0112).

The same parameter measured in degrees showed a moderate asymmetry in all the groups. The differences between the groups were not statistically significant. The results also show the occurrence of asymmetry in the distance of the right and left shoulder blades in relation to the spine line in all the groups. The biggest difference was noticed in the volleyball training group. The differences between the groups were not statistically significant. The power of the test was good. The model showing the differences between the study groups in the distance of the shoulder blades in relation to the spine line was also analyzed. There were no statistically significant differences between the groups.

The analysis of the parameters describing the height and width of the waist showed that moderate asymmetries occurred in all the study groups. The biggest difference in the waist height occurred in the swimming group, and in the waist width in the biathlon/taekwondo group. The differences between the groups were not statistically significant. The detailed summary of data in the GLM model is presented in Table 5.

In the group of training and non-training children, no statistically significant differences were observed between the parameters describing the position of the pelvis. No significant asymmetries were observed in any of the studied groups. The angle of inclination and twisting of the posterior iliac spines measured in millimeters showed no asymmetry in either group. The mean values of the above-mentioned angles, measured in degrees, showed slight and moderate asymmetries in each group, regardless of the type of sport practiced. The greatest asymmetries in the angle defining the inclination of the posterior iliac spines occurred in the swimming group, and the greatest angle of twisting of the posterior iliac spines was in the biathlon/taekwondo group and in the volleyball group. The characteristics of the KNM and KSM parameters in the GLM model are presented in Table 6.

The Spearman’s correlation analysis between the parameters determining the body posture did not show any statistically significant relationships between the position of the shoulders and shoulder blades and the position of the pelvis in the tested people.

## 4. Discussion

The aim of the study was to assess the body posture in children practicing selected sports disciplines in the public primary sports-profile schools or in children training in sports clubs in Warsaw and in children attending public primary schools in Warsaw, which constitute a control group. Types of postural disorders and their frequency in the studied groups, as well as the level of physical activity in training and non-training children, were also assessed. Somatic parameters of all the groups and correlations between various variables and parameters characterizing the body posture were also compared.

In school-age children and adolescents, changes in body posture are currently a significant problem. Civilizational progress, the development of technology, and change of lifestyle as well as practicing various sports disciplines can affect the developing figure of a child in different ways. When analyzing the current literature we did not find a clear answer to the question of the impact of training various sports on the body posture of school-aged children. Many publications pointed to the positive impact of physical activity and practicing various sports on body posture [17,18], whereas some other publications drew attention to the fact that asymmetric physical activity can have a negative impact on various elements of the body posture [19,20,21,22,23,27,30,36,37,38].

The author was interested in this age group because of the fact that the children’s growth spurt ends only after the age of 13 years [39]. It is then that the greatest plasticity and susceptibility to disorders within the musculoskeletal system occurs. Understanding the relationship between practicing various sports disciplines and body posture may help to maintain the correct body posture thanks to the training process and adequate prophylaxis. It is especially important because posture defects in the body of school-aged athletes may develop even within a few weeks [13]. If detected early enough they can be quickly corrected, whereas if untreated they can have a negative impact on the training process and on the child’s sports performance. Another argument is the fact that the given age means that the child attends grades 4-6 in a primary school. The choice of these classes was related to the start of practicing the selected sports discipline from the fourth grade of primary school. Earlier, children train in sports or engage in recreational physical activity on their own.

On the basis of our own research, in all the examined groups, asymmetry of moderate intensity was observed in the parameters characterizing the position of the shoulders, shoulder blades, and the waist triangle both in training and non-training people. In all the groups there was no asymmetry in the parameters determining the depth of the shoulder blades, as well as in the position of the pelvis in the frontal and transverse planes. No significant intensity of asymmetry was noted in any of the examined parameters.

Mrozkowiak et al. [40] compared in their work the habitual posture of athletes practicing various sports disciplines by using the Moiré method. The study involved 18 men practicing volleyball (aged 20 to 35) and 15 men practicing football (aged 15 to 32). Training experience in volleyball was of 13 years, and in football 14 years. Volleyball players had a higher position of the left shoulder in relation to the right one, a higher position of the left waist triangle, and a smaller distance of the left shoulder blade to the spine in relation to the right shoulder blade. There was also a more frequent twisting of the pelvis to the left side. Football players were characterized mainly by a higher setting of the left waist triangle. There was also a slight elevation of the right shoulder and the right shoulder blade, and the position of the left shoulder blade was closer to the line of the spine. In this group of training people, pelvic twisting in the transverse plane to the left side also prevailed. The results of our own research did not show a statistically significant difference between pairs of groups in the studied parameters in people practicing volleyball and football. Therefore, it cannot be unequivocally determined whether the training of the above-mentioned sports disciplines had a negative impact on the parameters that characterize the body posture of the examined people.

The body posture of football players, using the Moiré phenomenon, was also analyzed by Grabara [37]. The study qualified 73 football-training children aged 11–14 and 78 non-training children of the same age. The training period was 2 years for people aged 11 and 4 years for people aged 13. For players aged 11, the number of training sessions was three times per week, whereas for players aged 13, the number of training sessions was five times. The results of the study showed that the only parameter showing significant statistical differences in all the age groups was the angle of the pelvis. The comparison of shoulder asymmetry also showed a more symmetrical arrangement of the shoulders in relation to each other in children training football versus their non-training peers. However, the differences were not statistically significant. On the basis of the above results, it can be assumed that training football three to five times per week had a positive effect on the body posture parameters in the frontal plane. In this case, however, it is difficult to draw clear conclusions that football training was the main factor in improving the body posture of children aged 11–14. Such an unequivocal conclusion cannot be reached by analyzing the results of our own research. Our results showed no asymmetry in the parameter describing the pelvic inclination angle in all the examined people, including football players. In all the groups, however, moderate asymmetry was found in the parameter characterizing the position of the shoulders. No statistically significant difference between the groups was found.

In another study, Grabara [36] by using the Moiré method compared the body posture of volleyball players and non-training people. She qualified for the study 104 players aged 14–16 and their 114 non-training peers. The training period ranged from 2 to 6 years. The right shoulder was ≥5 mm higher than the left shoulder in 37% of volleyball players and in 29% of non-training people. On the other hand, 40% of people training volleyball and 45% of non-training people did not show significant shoulder asymmetry (the difference in levels was below 5 mm). The right shoulder blade was set at a bigger distance from the spine than the left one in 46% of people training volleyball and in 39% of non-training people. The impact of asymmetric loading of the body and spine during the volleyball training did not negatively influence the body posture parameters in the frontal plane. In order to determine the improvement or deterioration of the body posture resulting from practicing individual sports, the same group of participants should be tested several times, at different time intervals. Such an analysis was made by Grabara [41] among volleyball players—13 girls and 19 boys were examined three times: at the age of 14, 15, and 16. The training period at the time of the first measurement did not exceed 5 months and there were 5 training sessions per week of 90 min each. The study showed in training people a statistically significant difference between the measurements only in the case of the angle of pelvic twisting to the right in the transverse plane. The average value of the angle of pelvic twisting during the first measurement was 5.60, and during the third measurement, it was 9.10. An increase in pelvic twisting also occurred in the control group in non-training people. In the volleyball training group, a greater distance of the right shoulder blade from the spine in relation to the left shoulder blade and a greater elevation of the right shoulder were also noted. Therefore, there is a need to introduce additional symmetrical, stabilization, and stretching exercises to volleyball training. On the basis of our research, it is difficult to say how the physical activity and the type of training influenced this in the study groups. More studies with a larger sample size are necessary to be able to determine these aspects, although it is a challenging task.

Another study determining the impact of training various sports disciplines on the body posture in children attending primary school was the work of Tomenko et al. [42]. The study involved 43 children who apart from attending physical education classes, also took part in extracurricular activities in basketball (17 boys) and taekwondo (14 boys and 12 girls). A total of 27 children (16 boys and 11 girls) did not choose any extracurricular physical activity. A positive effect of practicing taekwondo on the assessment of body posture was demonstrated. The number of children with correct posture in this group was 24.3% which was 10% higher than in the basketball training group and 11.4% higher than in non-training children. The results of our own research showed a statistically significant difference in the model describing the UB parameter, measured in millimeters. The group training biathlon and taekwondo showed a greater asymmetry of the above parameter than other groups. Children training in biathlon and taekwondo were within the lower norm of moderate asymmetry, whereas in the remaining groups, the average values of the UB parameter in the statistical model did not show asymmetry.

The influence of physical activity on body posture parameters in adolescents aged 14–16 was also studied by Mucha et al. [17]. The group of adolescents with increased physical activity showed a much better body posture in relation to the group of adolescents with average physical activity. The analysis showed symmetry in the setting of the shoulder blades in relation to the spine in 67% of people with increased physical activity. For comparison, in the group with average physical activity 40% of adolescents showed such a symmetry. Symmetrical setting of the shoulders, shoulder blades, and pelvis occurred in 27% of the group with increased physical activity and in 14% of the group with average physical activity.

The results of our own research did not allow us to clearly determine whether practicing various sports disciplines has a negative or positive effect on body posture. There were also no significant asymmetries (above 10 mm) in any of the tested parameters. Additionally, no significant differences between children practicing various sports and non-training children were indicated despite the insufficient number of physical activities per week in the control group. It can be assumed that the differences between the results of our own research and the results of other authors result from a different number of training sessions and different training experiences. The results of each study are not affected only by the physical activity and the type of sport practiced, but also by other disturbing factors such as age, leisure time, and extracurricular activity.

The somatic parameters differentiated the examined groups, including training and non-training groups, in terms of body height and weight. According to the specificity of given sports disciplines, the biggest body height was found in children practicing volleyball.

The limitation of this study was the inability to conduct the study twice in a group of training children, several years apart. This was due to organizational difficulties, changing the class by children, or the lack of written consent of parents for further research after a period of a few years. Examining children twice would make it possible to determine precisely what effect training in a given discipline has on body posture at different ages. Another limitation of this study was the inability to test a larger group of training children and to include in it a wider range of sports disciplines. Our research was carried out only in schools where principals and trainers gave their consent to conduct such research. Additionally, a number of the children’s parents refused to give their consent. Research had to be stopped due to the coronavirus pandemic and the closure of school facilities and the suspension of the training process in sports clubs. For this reason, another limitation of the research is the combination of biathlon and taekwondo training groups into one group and the differentiation of groups in terms of time and training load. The children attended the same class and had the same general physical development classes, but their specialization training was different, specific to the selected sports discipline. In the opinion of the authors of this study, the direction of future research should be (1) the expansion of the research group and independent examination of children practicing different sports disciplines and (2) the inclusion of a larger number of asymmetric sports disciplines into the study. Perhaps this would enable a more thorough understanding of the relationship between training asymmetric sports and body posture, which is particularly important in children at early school age. The analysis of the results of our study did not allow us to determine the positive or negative impact of training on body posture. Conducting research of a wider scope would allow for a better understanding of the above relationships which in turn would enable the refinement of the training plan and improve sports performance.

All children groups in our study are very heterogeneous among themselves in terms of age, height, weight, BMI, training duration, and training load. It was related to specific sport selection features. That is another important limitation of our study that could be found while determining factors in the obtained results.

## 5. Conclusions

Our work has led us to conclude that most of the examined people had a correct body posture in the sagittal plane, regardless of the type of sport practiced. In all the examined groups, the most common dysfunctions were asymmetries of moderate intensity in the frontal plane.

The results of our own research did not allow us to clearly state whether practicing different sports disciplines and different training loads had a negative or positive impact on body posture. The lack of asymmetry of high intensity in the groups of people practicing various sports disciplines, despite the fact that the given disciplines are asymmetric, may indicate that exercises during the training process are correctly selected.

## Figures and Tables

**Figure 1 ijerph-20-04855-f001:**
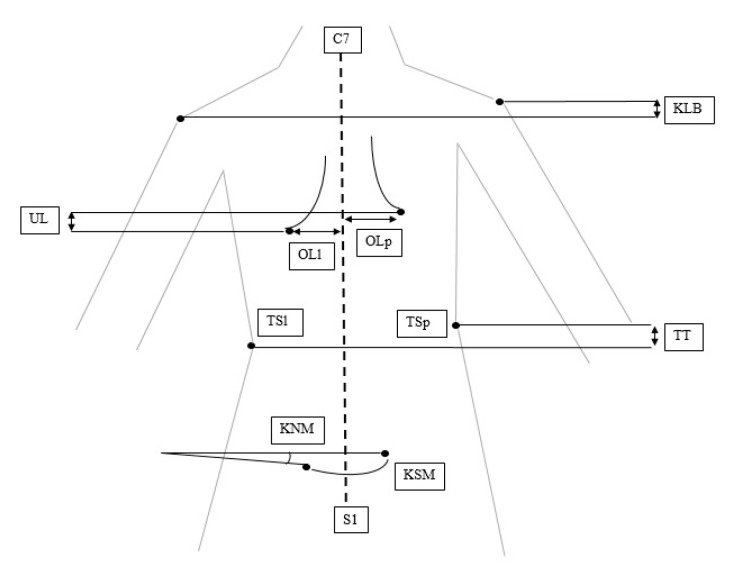
Listed body posture evaluation parameters. Source: own material.

**Table 1 ijerph-20-04855-t001:** Biometric data of the examined children.

		Group					
Parameter	Gender	Biathlon/ Taekwondo	Football	Volleyball	Swimming	Control	*p*
N [number of people]	F	29	0	44	32	28	-
	M	33	60	17	32	35	
	Total	62	60	61	64	63	
Age [years]	F	10.7 ± 1.3	-	11.9 ± 1.0	10.3 ± 0.7	11.2 ± 1.1	
	M	11.1 ± 1.3	11.0 ± 1.0	11.4 ± 1.2	10.4 ± 0.9	11.3 ± 1.2	
	Total	10.9 ± 1.3	11.0 ± 1.0	11.8 ± 1.1	10.3 ± 0.8	11.3 ± 1.1	*p* < 0.0001
Height of the body [cm]	F	145.4 ± 7.4	-	157.9 ± 11.2	143.7 ± 8.0	150.4 ± 9.3	
	M	148.5 ± 10.2	147.0 ± 8.6	150.2 ± 7.3	147.8 ± 8.0	151.7 ± 8.7	
	Total	147.0 ± 9.1	147.0 ± 8.6	155.8 ± 10.8	145.7 ± 8.2	151.1 ± 8.9	*p* < 0.0001
Weight of the body [kg]	F	37.1 ± 6.3	-	45.8 ± 11.5	35.2 ± 7.7	42.9 ± 8.1	
	M	39.6 ± 8.2	37.0 ± 8.3	37.1 ± 5.9	38.2 ± 6.7	43.9 ± 10.8	
	Total	38.4 ± 7.4	37.0 ± 8.3	43.4 ± 10.9	36.7 ± 7.3	43.4 ± 9.6	*p* < 0.0001
BMI [kg/m^2^]	F	17.5 ± 1.9	-	18.1 ± 2.6	16.9 ± 2.2	18.8 ± 2.3	
	M	17.8 ± 2.2	16.9 ± 2.5	16.4 ± 1.4	17.4 ± 2.3	18.9 ± 3.6	
	Total	17.6 ± 2.0	16.9 ± 2.5	17.6 ± 2.4	17.2 ± 2.2	18.9 ± 3.1	0.0007

**Table 2 ijerph-20-04855-t002:** Training loads in the studied groups.

Group	Descriptive Statistics ($\bar{x}$ ± SD)	
	Training Duration [Months]	Training Load [Hours Per Week]
Biathlon/taekwondo	27.6 ± 20.9	8.2 ± 2.9
Football	53.1 ± 22.2	10.7 ± 4.7
Volleyball	23.7 ± 16.8	8.6 ± 2.7
Swimming	41.4 ± 19.0	8.4 ± 3.1
*p*	<0.0001	0.0013

**Table 3 ijerph-20-04855-t003:** Characteristics of the KLB parameter.

Group	Descriptive Statistics ($\bar{x}$ ± SE)	
	KLB [mm]	KLB [Degrees]
Biathlon/taekwondo	7.24 ± 0.74	1.44 ± 0.15
Football	6.14 ± 0.60	1.29 ± 0.13
Volleyball	5.18 ± 0.60	1.07 ± 0.12
Swimming	6.14 ± 0.60	1.26 ± 0.12
Control	5.30 ± 0.62	1.07 ± 0.13
*p*	0.2828	0.2798

**Table 4 ijerph-20-04855-t004:** Characteristics of the UL, UB, and OL parameters.

Group	Descriptive Statistics ($\bar{x}$ ± SE)				
	UL [mm]	UL [Degrees]	UB [mm]	UB [Degrees]	OL [mm]
Biathlon/taekwondo	6.39 ± 0.70	2.70 ± 0.30	6.25 ± 0.76	2.70 ± 0.30	8.39 ± 0.98
Football	6.35 ± 0.77	2.68 ± 0.32	4.47 ± 0.44	2.68 ± 0.32	6.51 ± 0.97
Volleyball	6.03 ± 0.70	2.83 ± 0.33	4.21 ± 0.50	2.83 ± 0.33	8.70 ± 0.77
Swimming	5.67 ± 0.61	2.47 ± 0.28	3.68 ± 0.43	2.47 ± 0.28	7.05 ± 1.02
Control	6.92 ± 0.74	2.89 ± 0.29	4.23 ± 0.46	2.89 ± 0.29	8.53 ± 0.80
*p*	0.7763	0.9316	0.0049	0.0527	0.2755

**Table 5 ijerph-20-04855-t005:** Characteristics of the TT and TS parameters.

Group	Descriptive Statistics ($\bar{x}$ ± SE)	
	TT [mm]	TS [mm]
Biathlon/taekwondo	5.83 ± 0.66	9.07 ± 0.94
Football	6.26 ± 0.64	8.39 ± 1.05
Volleyball	5.00 ± 0.47	7.59 ± 0.99
Swimming	6.59 ± 0.61	6.35 ± 0.88
Control	5.25 ± 0.48	8.21 ± 0.72
*p*	0.4886	0.2239

**Table 6 ijerph-20-04855-t006:** Characteristics of the KNM and KSM parameters.

Group	Descriptive Statistics ($\bar{x}$ ± SE)			
	KNM [mm]	KNM [Degrees]	KSM [mm]	KSM [Degrees]
Biathlon/taekwondo	2.68 ± 0.37	1.71 ± 0.23	4.33 ± 0.44	2.80 ± 2.29
Football	2.83 ± 0.36	1.97 ± 0.24	3.46 ± 0.30	2.39 ± 0.21
Volleyball	2.71 ± 0.31	1.73 ± 0.20	4.39 ± 0.44	2.74 ± 0.24
Swimming	3.12 ± 0.30	2.03 ± 0.20	3.52 ± 0.41	2.28 ± 0.27
Control	2.84 ± 0.32	1.78 ± 0.20	4.41 ± 0.48	2.80 ± 0.30
*p*	0.9707	0.9007	0.2342	0.5020

## Data Availability

Raw data of this article are available upon request to the corresponding author.

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
