# Peer review of "Evaluation of Size of Trunk Asymmetry in Children Practicing Selected Sports Disciplines"

_ijerph, 2023, doi:10.3390/ijerph20064855_

Round 1
Reviewer 1 Report
ijerph-2233087_review
Title: Evaluation of size of a trunk asymmetry in children practicing selected sports disciplines
Comments for Authors
Dear authors,
I have carefully read your paper, which analysed and compared the body posture using the Moiré method in children practising sports disciplines such as biathlon/taekwondo, football, volleyball or swimming and non-training children.
In your results, no statistically significant relationships were observed between the groups for the position of the shoulders and shoulder blades and the position of the pelvis, in addition most of the people examined had a correct body posture in the sagittal plane, regardless of the type of sport practised, the most common dysfunctions were asymmetries of moderate intensity in the frontal plane. Therefore, your results did not clearly state whether practising different sports disciplines and different training loads had a negative or positive impact on the body posture.
I found some issues in introduction, methods, results, discussion and conclusion sections that should be addressed to improve the paper, in my opinion.
Specific comments:
Abstract: Page 1, lines 13-14: I suggest you change this sentence to the methods section.
Introduction
- Page 2, lines 47-48. This sentence does not contain any references; please add the references that are necessary to support this information.
- Page 2, lines 49-51. This paragraph does not contain any references; please add the references that are necessary to support this information.
- Page 2, line 58. In the discussion section, you briefly present the hypothesis-justification of your study. This is crucial for the understanding of the manuscript; however, you have not specified the hypothesis of your study in the introduction. I suggest that you briefly add this information before mentioning the objective of your intervention.
Methods
- Page 2, line 63. Please add information about your study design.
- Page 2, line 63. Why do you select these sports disciplines and not others? Please justify your answer. Add the necessary references.
- Page 2, lines 63-64. You obtained the participants from primary sports schools or sports club. How many different schools or clubs participated? Are they from the same cities, from the same country? How many participants belonged to each school or club? Please complete this information
- Page 2, line 69. How did you make sure that the participants in the control group did not perform any physical activity? How do you control the physical activity that each participant performs outside the primary sports schools or sports club. during the entire duration of the study? I consider that this is a very important parameter that could condition your results.
- Page 2, Table 1. This table shows the results of your study, I suggest you move it to the results section.
- Page 3, line 74: Please, could you add information about the period of time and places in which the data were recorded? How long did it take to complete the entire assessment? Who made the measurements? Were they the same evaluators for all the sports schools or sport clubs?
- Page 3, line 79, You mentioned in this line the inclusion criteria “age interval of the children was from 8.5 to 13.5 years…” This interval does not coincide with that indicated previously in page 2 line 70, please check this information and unify it.
- Page 3, line 90: Please add information about the tools used to assess the weight, height and BMI of your study participants. Add the necessary references.
- Page 3, line 90: Please, add references for the guidelines of the device’s manufacturer and for the Moiré method.
- Page 4, lines 107-111. I have many doubts about the statistical analysis. First of all, you do not mention how you have calculated the sample size for your study, this is crucial to support the validity of your results, please add this information in detail. You have not mentioned the statistical package or program that you have used to perform the statistical analysis, please add this information. How did you calculate the test power? Have you evaluated the normality of the variables? if so, add this information
Results
- Table 1: The results in Table 1 show that the groups in your study are very heterogeneous among themselves in terms of age, height, weight, BMI, training duration and training load. All p values are significant. The number of months from the moment the training started is also very different between the groups. the group of swimmers or football players have been training for almost 4 years compared to the group of taekwondo or volleyball players who have been training for an average of 2 years. These are a very important limitations of your study that could be found determining factors in the results you have obtained.
Tables 3, 4, 5, 6 I suggest you add information in these tables, you could add the p value when individually comparing each of the groups that practice sports with the control group, I think that in this way you could still obtain more complete results that would provide more information about this study.
Discussion
- Page 17, line 206. Grammar mistake “corrected”, please check it.
- Page 18, lines 249-250: there is a space left, check it.
- You adequately mention some limitations of the study. As I have previously commented in the methodology section, I consider that you must add more limitations about parameters that could condition your results. I suggest add limitations of your study.
- Page 18, lines 269-270. You mentioned that “It is impossible to say how the physical activity and the type of training influenced this in the study groups”, Are you sure of this statement? Please, I suggest you consider modifying it. Surely more studies with a larger sample size are necessary to be able to determine these aspects, although it is complicated, surely it will not be impossible, I encourage you to continue investigating in this regard.
Conclusions
- Page 19. Lines 329-332. You mention in your conclusion that the sports you have selected are asymmetric sports. I think this is a key concept for understanding your study, please add more information on this aspect that supports this statement in the introduction.
I hope that my comments could help to improve the paper.
Author Response
Thank you very much for all your valuable comments. All your suggestions - which we really appreciate - will certainly improve the quality of our article. Please find below the list of changes that we have made in our publication.
Reviewer's note: Abstract: Page 1, lines 13-14: I suggest you change this sentence to the methods section.
Answer: Reworded as suggested. Thank you
Abstract: (1) Background: The aim of the study was to assess the body posture in children practicing selected sports disciplines and to compare it to the body posture of non-training children. (2) Methods: 247 children practicing a selected discipline either in a primary sports schools or in a sports clubs constituted the study group. The control group was composed of 63 children that did not practice any sport. The study of the body posture by using the Moiré method allowed to assess the size of parameters determining the body posture. Selected parameters characterizing the position of the shoulders and shoulder blades, the waist triangle and the position of the posterior iliac spines were analyzed.
Introduction
Reviewer's note: Page 2, lines 47-48. This sentence does not contain any references; please add the references that are necessary to support this information.
Answer: References have been added
The spontaneous and accurate posture correction without help of a specialist is a rarely observed phenomenon [5].
Reviewer's note: Page 2, lines 49-51. This paragraph does not contain any references; please add the references that are necessary to support this information.
Answer: References have been added. Thank you
Reviewer's note: Page 2, line 58. In the discussion section, you briefly present the hypothesis justification of your study. This is crucial for the understanding of the manuscript; however, you have not specified the hypothesis of your study in the introduction. I suggest that you briefly add this information before mentioning the objective of your intervention.
Answer: Introduction has been reworded, Thank you
In this study we posed following questions: (1) what body posture disorders occur most frequently in the groups of training and non-training children? (2) how often do they occur? (3) what types of body posture disorders are characteristic to selected sports disciplines?
Methods
Reviewer's note: Page 2, line 63. Please add information about your study design.
Answer: Sentence reworded
This research is an observational, case-control study.
Reviewer's note: Page 2, line 63. Why do you select these sports disciplines and not others? Please justify your answer. Add the necessary references.
Answer: The description of the limitations of the study has been clarified
The limitation of this study was the inability to conduct the study twice in a group of training children, several years apart. This was due to organizational difficulties, changing the class by children or the lack of written consent of parents for further re-search after a period of a few years. Examining children twice would make it possible to determine precisely what effect training of a given discipline has on the body posture at different age. Another limitation of this study was the inability to test a larger group of training children and to include in it a wider range of sports disciplines. Our research was carried out only in schools where principals and trainers gave their consent to conduct such research. Additionally, a part of the children’s parents refused to give their consent. Research had to be stopped due to the coronavirus pandemic and the closure of school facilities and suspension of training process in sports clubs. . For this reason, another limitation of the research is the combination of biathlon and taekwondo train-ing groups into one group and differentiation of groups in terms of time and training load. The children attended the same class and had the same general physical devel-opment classes, but their specialization training was different, specific to the selected sports discipline. In the opinion of the authors of this study the direction of future re-search should be: (1) the expansion of the research group and independent examination of children practicing different sports disciplines and (2) inclusion of a larger number of asymmetric sports disciplines into the study. Perhaps this would enable a more thor-ough understanding of the relationship between training asymmetric sports and body posture, which is particularly important in children at early school age. The analysis of the results of our study did not allow to determine the positive or negative impact of training on body posture. Conducting research of a wider scope would allow to under-stand better the above relationships which in turn would enable to refine the training plan and improve sports performance.
All children groups in our study are very heterogeneous among themselves in terms of age, height, weight, BMI, training duration and training load. It was related to specific sport selection features. That is another important limitations of our study that could be found while determining factors in the obtained results.
Reviewer's note: Page 2, lines 63-64. You obtained the participants from primary sports schools or sports club. How many different schools or clubs participated? Are they from the same cities, from the same country? How many participants belonged to each school or club? Please complete this information
Answer: Reworded as suggested
All the participants attended primary schools and sports clubs in Warsaw, Poland. The study was conducted in four primary schools and two sports clubs.
Reviewer's note: Page 2, line 69. How did you make sure that the participants in the control group did not perform any physical activity? How do you control the physical activity that each participant performs outside the primary sports schools or sports club. during the entire duration of the study? I consider that this is a very important parameter that could condition your results.
Answer: Reworded as suggested
The number and type of physical activity of the participants in the control group was specified by using an original questionnaire.
Reviewer's note: Page 2, Table 1. This table shows the results of your study, I suggest you move it to the results section.
Answer: Reworded as suggested
The table has been moved to the “Results” section
Reviewer's note: Page 3, line 74: Please, could you add information about the period of time and places in which the data were recorded? How long did it take to complete the entire assessment? Who made the measurements? Were they the same evaluators for all the sports schools or sport clubs?
Answer: Information has been added as suggested
The study was conducted in 2017-2020. The tests were carried out during morning school hours or at a date which was pre-determined with the coaches of sports clubs "Sparta" and "Escola Varsovia". The collection of results was interrupted by the
COVID-19 pandemic, which led to the closure of schools and interruption of training process in sports clubs. The data was collected by one researcher (NTG) who, after completing the methodological part, processed the results in accordance with the recommendation of the device’s manufacturer [37].
[37] Åšwierc, A. Komputerowa diagnostyka postawy ciaÅ‚a – instrukcja obsÅ‚ugi. 2006 Czernica WrocÅ‚awska.
Reviewer's note: Page 3, line 79, You mentioned in this line the inclusion criteria “age interval of the children was from 8.5 to 13.5 years…” This interval does not coincide with that indicated previously in page 2 line 70, please check this information and unify it.
Answer: Sentence reworded and removed to the “Results” section
Reviewer's note: Page 3, line 90: Please add information about the tools used to assess the weight, height and BMI of your study participants. Add the necessary references.
Answer: Information has been added as suggested
Anthropometric measurements were done and anthropological indicators were calculated for each participant of the study. Body weight was determined with an accuracy of 0.1 kg by using an electronic scale. Body height was measured with an anthropometer with an accuracy of 1 cm. BMI was calculated for each child by using data on weight and height. BMI percentage charts were calculated based on the OLAF and OLA project carried out by the "Pomnik – Centrum Zdrowia Dziecka” Institute in the years 2007-2012 [38].
Reviewer's note: Page 3, line 90: Please, add references for the guidelines of the device’s manufacturer and for the Moiré method.
Answer: Reference added as suggested
In order to evaluate the body posture we used the Moiré method which is in line with the guidelines of the device’s manufacturer [37].
[37] Åšwierc, A. Komputerowa diagnostyka postawy ciaÅ‚a – instrukcja obsÅ‚ugi. 2006 Czernica WrocÅ‚awska.
Reviewer's note: Page 4, lines 107-111. I have many doubts about the statistical analysis. First of all, you do not mention how you have calculated the sample size for your study, this is crucial to support the validity of your results, please add this information in detail. You have not mentioned the statistical package or program that you have used to perform the statistical analysis, please add this information. How did you calculate the test power? Have you evaluated the normality of the variables? if so, add this information
Answer: Reworded as suggested
Simple descriptive statistics were initially calculated. For qualitative variables, the χ2 test or the Fisher's exact test was used, depending on the sample size. For quantitative variables, the hypothesis of equality of distributions was tested with the Kruskal-Wallis test. Spearman's correlations were used to examine the relationships between the studied variables.
Final analysis of the results was based on multidimensional statistical GLM (Generalized Linear Models) models. According to the Akaike Information Criterion (AIC), the most appropriate model parameters were selected.
To assess the repeatability of the measurements, repeated observations of an independent group of patients were analyzed and then the ICC coefficient was calculated in a random effects ANOVA model.
The calculations were made in the SAS package (SAS/STAT rel. 15.1). The value of p=0.05 was assumed as a significant level of confidence. In calculations that were statistically significant, the power of the test was shown to be valid.
Results
Reviewer’s note: Table 1: The results in Table 1 show that the groups in your study are very heterogeneous among themselves in terms of age, height, weight, BMI, training duration and training load. All p values are significant. The number of months from the moment the training started is also very different between the groups. the group of swimmers or football players have been training for almost 4 years compared to the group of taekwondo or volleyball players who have been training for an average of 2 years. These are a very important limitations of your study that could be found determining factors in the results you have obtained.
Answer: “Limitation” section reworded
Reviewer’s note: Tables 3, 4, 5, 6 I suggest you add information in these tables, you could add the p value when individually comparing each of the groups that practice sports with the control group, I think that in this way you could still obtain more complete results that would provide more information about this study.
Answer: Results section in the abstract has been reworded
Results: Differences in selected parameters were not statistically significant in all parameters except the model describing the values ​​determining the depth of the shoulder blades that were measured in millimeters between the groups.
Discussion
Reviewer’s note: Page 17, line 206. Grammar mistake “corrected”, please check it.
Reviewer’s note: Page 18, lines 249-250: there is a space left, check it
Answer: Mistakes have been corrected
Reviewers’s note: You adequately mention some limitations of the study. As I have previously commented in the methodology section, I consider that you must add more limitations about parameters that could condition your results. I suggest add limitations of your study.
Answer: Limitations have been added. Thank you for your suggestion.
Reviewer’s note: Page 18, lines 269-270. You mentioned that “It is impossible to say how the physical activity and the type of training influenced this in the study groups”, Are you sure of this statement? Please, I suggest you consider modifying it. Surely more studies with a larger sample size are necessary to be able to determine these aspects, although it is complicated, surely it will not be impossible, I encourage you to continue investigating in this regard.
Answer: Sentence has been reworded
On the basis of our research it is difficult to say how the physical activity and the type of training influenced this in the study groups. More studies with a larger sample size are necessary to be able to determine these aspects, although it is a challenging task.
Conclusions
Reviewer’s note: Page 19. Lines 329-332. You mention in your conclusion that the sports you have selected are asymmetric sports. I think this is a key concept for understanding your study, please add more information on this aspect that supports this statement in the introduction.
Answer: Introduction reworded
Many publications have confirmed the positive effect of practicing sports on body posture [17,18]. On the other hand, some researchers have reported negative consequences of asymmetric physical activity on various elements of body posture [19,20,21,22,23].
Reviewer 2 Report
In this paper, the authors investigated asymmetry in children practicing selected sports disciplines. This research attempts to show that if practicing different sports disciplines could had influences on the body posture of children. This topic is useful, and the data collected in this study is also great. However, this paper's quality needs to be improved. Some questions and comments are the following:
1. Figure 1 has red underlines, please use save as figure instead of just capturing the screenshot directly.
2. Please also give the limitations to this study. For example, please discuss why there is no significant negative or positive impact on body posture by practicing different sports disciplines and different training loads and discuss what if the significant impact could be revealed by improving the explement adopted in this study.
3. The conclusion part seems no need for itemization. If need, the authors could write like this:
In this study, we xxxx. Our study reveals that xxxx. In summary, our conclusions could be listed as follows:
1. XXXXX
2. XXXXX
4. Minor mistakes also could be found such as, Line 327 practising -> practicing. Please check them.
Author Response
Thank you very much for all your valuable comments. All your suggestions - which we really appreciate - will certainly improve the quality of our article. Please find below the list of changes that we have made in our publication.
Reviewer's note: Figure 1 has red underlines, please use save as figure instead of just capturing the screenshot directly.
Answer: Figure 1 has been corrected as indicated. Thank you
Reviewer's note: Please also give the limitations to this study. For example, please discuss why there is no significant negative or positive impact on body posture by practicing different sports disciplines and different training loads and discuss what if the significant impact could be revealed by improving the explement adopted in this study.
Answer: Research limitation was reworded
The limitation of this study was the inability to conduct the study twice in a group of training children, several years apart. This was due to organizational difficulties, changing the class by children or the lack of written consent of parents for further research after a period of a few years. Examining children twice would make it possible to determine precisely what effect training of a given discipline has on the body posture at different age. Another limitation of this study was the inability to test a larger group of training children and to include in it a wider range of sports disciplines. Our research was carried out only in schools where principals and trainers gave their consent to conduct such research. Additionally, a part of the children’s parents refused to give their consent. Research had to be stopped due to the coronavirus pandemic and the closure of school facilities and suspension of training process in sports clubs. . For this reason, another limitation of the research is the combination of biathlon and taekwondo training groups into one group and differentiation of groups in terms of time and training load. The children attended the same class and had the same general physical development classes, but their specialization training was different, specific to the selected sports discipline. In the opinion of the authors of this study the direction of future research should be: (1) the expansion of the research group and independent examination of children practicing different sports disciplines and (2) inclusion of a larger number of asymmetric sports disciplines into the study. Perhaps this would enable a more thorough understanding of the relationship between training asymmetric sports and body posture, which is particularly important in children at early school age. The analysis of the results of our study did not allow to determine the positive or negative impact of training on body posture. Conducting research of a wider scope would allow to under-stand better the above relationships which in turn would enable to refine the training plan and improve sports performance.
All children groups in our study are very heterogeneous among themselves in terms of age, height, weight, BMI, training duration and training load. It was related to specific sport selection features. That is another important limitations of our study that could be found while determining factors in the obtained results.
Reviewer's note: The conclusion part seems no need for itemization.
Answer: The “Conclusions” part has been redrafted as indicated
Our work has led us to conclude that most of the examined persons had a correct body posture in the sagittal plane, regardless of the type of sport practiced. In all the examined groups, the most common dysfunctions were asymmetries of moderate intensity in the frontal plane.
The results of our own research did not allow to clearly state whether practicing different sports disciplines and different training loads had a negative or positive impact on the body posture. The lack of asymmetry of high intensity in the groups of persons practicing various sports disciplines, despite the fact that the given disciplines are asymmetric, may indicate that exercises during the training process are correctly selected.
Reviewer's note: Minor mistakes also could be found such as, Line 327 practising -> practicing. Please check them.
Answer: Minor mistakes such as “practising” have been corrected. Thank you
Reviewer 3 Report
The authors have presented an interesting study on the posture of adolescents. Of particular interest is the large group across different sports with a control group. However, in addition to the interesting results, a few comments regarding the content stood out.
Introduction
The introduction is very short. The problem presentation and introduction of the reader should be revised by the authors. Especially with regard to the discussion of the own results the authors should take care that the introduction can not be completely decoupled from the discussion.
Method
In my opinion it does not make sense to put biathlon/taekwondo into one group even if the students are in the same class. The sports are very different in terms of the requirements profile.
It should be better described how the measurements are collected. For readers without deeper knowledge of the subject, this is not comprehensible.
Also the way of statistical calculation should be described in more detail. Also used packages and software versions are important.
Results
How to display a boxplot for UB only?
Should P be the value for the interclass correlation described in the methods.
The statistic does not seem to be described correctly. Please revise this. If an ICC was actually calculated here, then it must also be indicated which icc was calculated!
Discussion
The paragraphwise presentation of other studies should not be part of the discussion. Here it would be desirable if the own results are discussed concisely in the context of other published studies. It would be useful if the majority of publications presented in the introduction were included in the discussion.
Conclution
This is written very concisely and gives the reader a good impression of the core statements.
Author Response
Thank you very much for all your valuable comments. All your suggestions - which we really appreciate - will certainly improve the quality of our article. Please find below the list of changes that we have made in our publication.
Reviewer's note: The problem presentation and introduction of the reader should be revised by the authors. Especially with regard to the discussion of the own results the authors should take care that the introduction can not be completely decoupled from the discussion.
Answer: Introduction has been revised
In this study we posed following questions: (1) what body posture disorders occur most frequently in the groups of training and non-training children? (2) how often do they occur? (3) what types of body posture disorders are characteristic to selected sports disciplines?
And also:
Many publications have confirmed the positive effect of practicing sports on body posture [17,18]. On the other hand, some researchers have reported negative consequences of asymmetric physical activity on various elements of body posture [19,20,21,22,23].
The current state of knowledge validates also that there is an increase in the percentage of postural defects in children and adolescents in the last decade in spite of physical activity practiced by them [3,13,14,24,25,26].Sports disciplines that increase the risk of postural asymmetry include: among others volleyball, team sports, and combat sports [15,16,20]. There are many studies confirming the incorrect type of posture in children prac-ticing volleyball. VaÅ™eková et al. [22] observed asymmetry in the position of the shoul-der, scapula and posterior superior iliac spine on the dominant side in professional players. KuczyÅ„ski et al. [28] presented a different theory showing that practicing vol-leyball results in better body posture stability and a different mode of automatic posture control in training children compared to the control group.In recent years, the impact of martial arts training on body posture has also been analyzed. WaÅ‚aszek et al. [31] assessed body posture changes in a group of 6-year-olds practicing judo.. Sunghak et al. [32] also showed a positive effect of martial arts on body posture correction.
Reviewer's note: In my opinion it does not make sense to put biathlon/taekwondo into one group even if the students are in the same class. The sports are very different in terms of the requirements profile.
Answer: Research limitation was rewordedThe limitation of this study was the inability to conduct the study twice in a group of training children, several years apart. This was due to organizational difficulties, changing the class by children or the lack of written consent of parents for further research after a period of a few years. Examining children twice would make it possible to determine precisely what effect training of a given discipline has on the body posture at different age. Another limitation of this study was the inability to test a larger group of training children and to include in it a wider range of sports disciplines. Our research was carried out only in schools where principals and trainers gave their consent to conduct such research. Additionally, a part of the children’s parents refused to give their consent. Research had to be stopped due to the coronavirus pandemic and the closure of school facilities and suspension of training process in sports clubs. . For this reason, another limitation of the research is the combination of biathlon and taekwondo training groups into one group and differentiation of groups in terms of time and training load. The children attended the same class and had the same general physical development classes, but their specialization training was different, specific to the selected sports discipline. In the opinion of the authors of this study the direction of future research should be: (1) the expansion of the research group and independent examination of children practicing different sports disciplines and (2) inclusion of a larger number of asymmetric sports disciplines into the study. Perhaps this would enable a more thorough understanding of the relationship between training asymmetric sports and body posture, which is particularly important in children at early school age. The analysis of the results of our study did not allow to determine the positive or negative impact of training on body posture. Conducting research of a wider scope would allow to under-stand better the above relationships which in turn would enable to refine the training plan and improve sports performance.All children groups in our study are very heterogeneous among themselves in terms of age, height, weight, BMI, training duration and training load. It was related to specific sport selection features. That is another important limitations of our study that could be found while determining factors in the obtained results
Reviewer's note: It should be better described how the measurements are collected. For readers without deeper knowledge of the subject, this is not comprehensible.
Answer: Information has been added as suggestedThe parameters listed above can be calculated by marking the appropriate points on the body of the examined child. In this study, the spinous processes of all vertebrae of the spine, the cervicothoracic and thoracolumbar junctions, the peak of kyphosis and lordosis, and the sacrum were marked. In addition, the inferior angle of the scapula, the posterior iliac spines, and the greatest depression in the waist were marked. The test was conducted in a darkened room, and the subject was positioned perpendicularly and 2.6 meters from the equipment. The photograms were evaluated after completion of the study, without the participation of the subject.
Reviewer’s note: Also the way of statistical calculation should be described in more detail. Also used packages and software versions are important.
Answer: The description of statistical calculations has been redraftedSimple descriptive statistics were initially calculated. For qualitative variables, the χ2 test or the Fisher's exact test was used, depending on the sample size. For quantitative variables, the hypothesis of equality of distributions was tested with the Krus-kal-Wallis test. Spearman's correlations were used to examine the relationships between the studied variables.Final analysis of the results was based on multidimensional statistical GLM (Generalized Linear Models) models. According to the Akaike Information Criterion (AIC), the most appropriate model parameters were selected. To assess the repeatability of the measurements, repeated observations of an in-dependent group of patients were analyzed and then the ICC coefficient was calculated in a random effects ANOVA model.The calculations were made in the SAS package (SAS/STAT rel. 15.1). The value of p=0.05 was assumed as a significant level of confidence. In calculations that were statistically significant, the power of the test was shown to be valid.
Reviewer's note: How to display a boxplot for UB only?
Answer: Boxplot has been changed and information has been added.
The model describing the values determining the depth of the shoulder blades that were measured in millimeters showed statistically significant differences between the examined groups. The statistical model showed a difference between the biathlon and taekwondo group and control group (p=0,0067) and swimming group (p=0,0112).
Reviewer's note: Should P be the value for the interclass correlation described in the methods.
The statistic does not seem to be described correctly. Please revise this. If an ICC was actually calculated here, then it must also be indicated which icc was calculated!
Answer: The description of statistical calculations has been redrafted
Reviewer's note: The paragraphwise presentation of other studies should not be part of the discussion. Here it would be desirable if the own results are discussed concisely in the context of other published studies. It would be useful if the majority of publications presented in the introduction were included in the discussion.
Answer: Introduction and discussion section have been reworded. Thank you very much
Round 2
Reviewer 1 Report
ijerph-2233087_review_R2
Title: Evaluation of size of a trunk asymmetry in children practicing selected sports disciplines
Comments for Authors
Dear authors,
I was glad to have the opportunity to review the new version of your manuscript, which analysed and compared the body posture using the Moiré method in children practising sports disciplines such as biathlon/taekwondo, football, volleyball or swimming and non-training children.
In my opinion, you have responded positively to the suggestions for improvement made, you have expanded the information required in the introduction and methods sections, you have made the modifications indicated in the results section, and reformulated the hypothesis and limitations.
I believe that all these modifications have improved the quality of this manuscript.
Therefore, I congratulate you on your great effort and the work you have done.
Author Response
Thank you very much for all suggestions and concerns.
All modifications have improved the quality of this manuscript.
Thank you very much for congratulations ?
Reviewer 2 Report
The authors have responded to my concerns.
Author Response
Thank you very much for all suggestions and concerns.
All modifications have improved the quality of this manuscript.
Thank you very much for acceptation.
Reviewer 3 Report
the comments from the first review round were answered satisfactorily. The paper can now be released.
Author Response

(The authors gave the same response as above.)
